# Optimal Ways of Unloading and Loading Operations under Arctic Conditions

**Marat Eseev and Dmitry Makarov \***

Higher School of Natural Sciences and Technologies, Northern (Arctic) Federal University, nab. Severnoi Dviny 17, 163002 Arkhangelsk, Russia; m.eseev@narfu.ru
**\*** Correspondence: makarovd0608@yandex.ru

**Abstract:** Usually, loading and unloading of cargo ships takes place in ports that are equipped with the infrastructure necessary to carry out such operations. In the Arctic, often a helicopter is the only way to get the cargo to the right place. Finding the optimal geographic location for unloading a ship using helicopters is an important task. It is necessary to create a support system for making the right decisions in such situations. Mathematical modeling has been used to find the geographical location that ensures the most favorable and quickest delivery of cargo from a vessel to its destination, using a helicopter. A criterion has also been found in which the search for the optimum point is a more rational way of unloading the vessel compared to other discharge options. The maps of the economic benefits of loading and unloading operations in this model have been developed. Using the example of the developed model, it is shown that during the transportation of goods in Ob Bay, significant economic and temporary advantages can be obtained. The developed model can be extended to the case of cargo delivery not only in the Arctic conditions, but also where the transport infrastructure is insufficiently developed.

**Keywords:** arctic; nothern sea route; decision support system; loading and unloading operations; ship; helicopter

## 1. Introduction

The development of the Arctic zone requires the solution of a whole range of problems; one of the most important challenges is the transportation of essential cargo to Arctic territories along the Northern Sea Route. Indeed, the route for the transport of goods between East Asia and Europe, using the Northern Sea Route, is approximately 14,000 km, compared with an alternative route through the Suez Canal of least 23,000 km. (see Figure 1). Despite the fact that the Northern Sea Route is the shortest, it is the most difficult, because Arctic conditions have a significant impact on transportation. Despite this, the Northern Sea Route is already being actively used for cargo transportation [1], and the melting of Arctic ice due to climate change is likely to lead to an increase in the turnover and profitability of this route [2–5].

For example, in August 2017 the tanker *Christophe de Margerie* became the first ship to sail the Northern Sea Route without the use of icebreaker wiring. The total duration of the journey from Hammerfest (Norway) to Poren (South Korea), using the Northern Sea Route, was 22 days, which is almost 30% less time than would be required for the traditional southern route through the Suez Canal. The outcome of this voyage was to reconfirm the economic efficiency of the use of the Northern Sea Route for the transit of large-capacity vessels, and prospects for more intensive use of the Northern Sea Route are currently being actively discussed [6–10]. In January 2021, a Yamalmax-class LNG carrier for the first time made an independent transition (without icebreaker escort) from the port of Sabetta along the Northern Sea Route to the east and reached the Bering Strait. At the same time, cargo was delivered from the Yamal LNG plant to the east; the average cruising speed of the *Christophe de Margerie* was 9.5 knots. The creation and management of transport infrastructure requires consideration

of the important natural and economic features inherent in this region can be enhanced by the use of mathematical modeling and traffic forecasting [11,12]. The constant growth of sea container traffic [13] makes it necessary to thoroughly analyze the problems at the strategic, tactical, and operational planning levels. At each level, optimization of the process is required, including selecting the way of loading and unloading the vessel, taking into account the technical capabilities of the port [14]. At the same time, it is important to determine the optimal route of vessels depending on the needs of customers, take into account many parameters, for example, the depths at the berths [15]. Preliminary mathematical modeling, taking into account the location of delivery points, helps to reduce time and financial costs [16]. However, original approaches and methods of mathematical modeling are needed in the search for solutions to transport problems. Standard methods of addressing transport problems in the Arctic region cannot be used, because there are limitations specific to this region, in particular the poor development of the transport network, including insufficient ports and berths capable of accepting the necessary cargo. Other restrictions may be because of ice and weather conditions, insufficient depth for passage of vessels and seasonality of navigation. The solution of the transport problem using classical formulation was solved in the 1940s by the Soviet mathematician and Nobel laureate in economics L.V. Kantorovich [17]. Since then, progress in the field of linear programming, the creation of banks and databases and the development of computer calculations have allowed the design of intelligent systems that manage traffic flows. However, the increase of data on Arctic transportation and the requirement for case-specific solutions to transport problems have driven the search for new mathematical tools and methods, and research in this area is now relevant and in demand both from a fundamental and applied point of view (see, for example, [8,18]). Considering the Northern Sea Route, usually, the effectiveness of its use is analyzed. For example, in the works [19–22], the economic efficiency of the Northern Sea Route is analyzed, where it is shown that this path may be preferable to other ways for certain delivery methods. The analysis is mainly determined by the dependence of the cost of delivery of goods from the traversed path in difficult Arctic conditions. At the same time, there is a great need for the development of methods of loading and unloading operations in the Arctic conditions. Along with large-scale analysis and mathematical modeling of macro flows of cargo during transportation in the Arctic, it is necessary to study the key operations for the delivery of goods under a range of conditions within the region, including in extreme situations. In this paper, we present an original method for solving the transport problem of loading and unloading vessels at several port points with the aim of selecting the optimal method and location of vessels when using a helicopter to move their cargo (see Figure 2). The use of helicopters, in Arctic conditions, is a well-known and long-used method of cargo delivery. The effectiveness of the use of helicopters in various situations is an urgent and important task [23].

The helicopter can be on board the ship or in relative proximity (hundreds of kilometers) from the unloading point. In the calculation, the economic feasibility, the time costs and the capability of the vessel to approach the shore for unloading are taken into account. It should be added that this way of delivering essential cargo has been practiced for a long time on the coast of the Northern Sea Route and through the Siberian rivers of the Ob, Yenisei and Lena into the interior of Siberia. Such deliveries always raise the question of whether it is expedient to search for the optimum unloading point of the vessel and the location of such a point. Usually this problem is solved very roughly (by evaluation methods), without involving mathematical modeling. Such appraisal approaches usually result in large errors which entail financial or time costs when delivering goods. That is why mathematical modeling of the processes of unloading and loading of ships is important. A particular problem is delivery to hard-to-reach hydrometeorological stations, for the majority of which sea vessels are the only available option for supplying them with food and fuel and changing personnel. The lack of suitably equipped berths at the ports combined with shallow water in the coastal zone and the condition of the coasts make it impossible to unload vessels ashore in most places. For these reasons, helicopters are

used on board ships, as well as for reloading to small ships and pontoons. In addition, when certain cargoes need to be delivered to specific points, the problem of the location of the cargo ship in the sea area may arise, so that again the use of helicopters often proves the most effective method in terms of cost and/or time. It is these problems that will be considered and solved in this work. The very problem of finding the optimal location of the object of delivery of cargo in several points, taking into account possible limitations, is in demand in various application areas. We focused on this task because it has practical application in a real situation. The Ob area of the Ob Bay is an actively developing transport highway in connection with the development of the South-Tambeyskoye oil and gas field on the Yamal Peninsula. There is a need to deliver goods to the Ob Bay region to the settlements: Sjojaha, Tambey, Antipayuta, Yamburg. At these points there are no equipped berths for the complete unloading of ships, therefore unloading is carried out by a helicopter, which is on the ship. Usually such a ship departs from Arkhangelsk. There is always the problem of choosing a point for unloading a vessel using a helicopter. However, geographic localization can be any, not limited to the considered example. The general approach makes it possible to determine from the available data the optimal positions and methods of moving goods in a similar situation anywhere in the world. You can change the carrier—the means of delivery, goods, consumers. A further situation is considered: the transportation of petroleum products along the Northern Sea Route during which there are environmental risks associated with potential oil spills in the seas of the Arctic zone. In the event of such an ecological catastrophe, it would be essential to deliver the necessary cargoes and means of minimizing the consequences a spill of oil products to certain points as quickly as possible. The model discussed here can be used in such a situation to enable swift and sound decision-making.

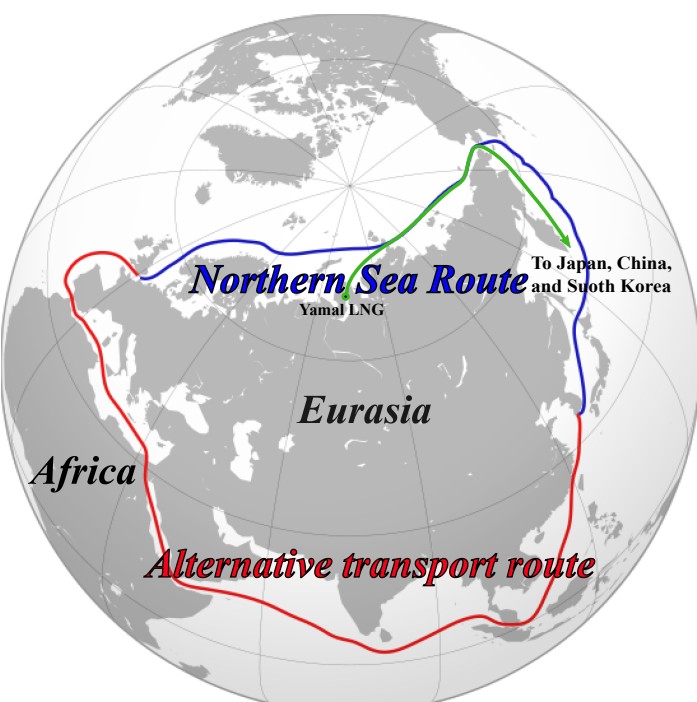

**Figure 1.** A map showing the transportation of goods by sea between East Asia and Europe along the Northern Sea Route (blue line) the Suez Canal (red line). In 2021, a Yamalmax-class LNG tanker made its first independent voyage (without icebreaker escort) with cargo from the port of Sabetta along the Northern Sea Route to the east (green line).

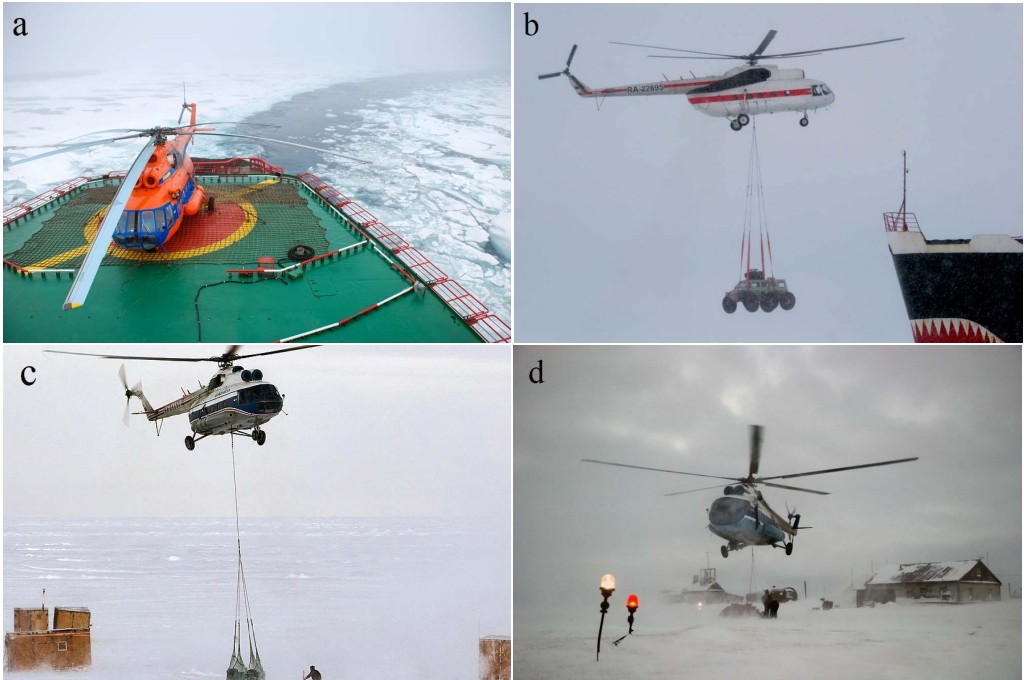

**Figure 2.** Use of a helicopter to unload a vessel: (**a**) is a helicopter on a cargo ship; (**b**) is the unloading from a vessel using a helicopter; (**c**) is the delivery from a payload ship for polar explorers; (**d**) is the delivery from a payload ship for residential and shift camps (Data in the free database https://unsplash.com, accessed on 2 February 2021).

## 2. Methods

Consider the problem of finding geographic locations for the most profitable transportation of cargo by ships and helicopters to points of unloading. In this task, there are two factors that need to be taken into account: the first is the cost or time spent during the transportation of cargo by helicopters from where the vessels have stopped; the secondt is the cost or time spent during transportation from the place of departure of vessels to the place of their unloading by helicopters. By definition of the task, the places of unloading of vessels are limited to a certain geographical area (for example, due to the coastline, shallow water, ice conditions in the coastal zone, etc.). We will assume that each ship unloads one helicopter (as is usually the case), which can be on board the unloaded vessel, or be located near the unloading points (a helicopter base).

Let us consider the first of the above factors, i.e., unloading cargo by helicopter. This task is related to the transport task [17,24–28], where $p$ production points are ships with cargo, and $n$ points of consumption are geographically fixed places. In the problem under consideration, the following data are needed: $a_i$ is the production volume at $i$ point, (one production point is a ship with cargo); $b_j$ is the volume of consumption at $j$ point; $c_{i,j}$ is the cost of transporting a unit of product from the ship with the number $i$ to unloading at point $j$ using the chosen route. The total production of $\sum_{i=1}^{p} a_i = \sum_{j=1}^{n} b_j$ is equal to the total consumption (all necessary cargo must be taken out of the ship). If $w_{i,j}$ is the transported volume (in units of mass) from the ship with the number $i$ to the unloading point $j$, then $\sum_{j=1}^{n} w_{i,j} = a_i$ and $\sum_{i=1}^{p} w_{i,j} = b_j$. In order to find the optimum way of unloading, we must consider the cost function $z_1 = \sum_{i,j=1}^{p,n} c_{i,j} w_{i,j}$. Since, for our problem, it is natural to assume that $c_{i,j} = S_{i,j}/m$, where $S_{i,j}$ is the cost, and $m$ is the average mass of the transported cargo per helicopter trip, we will assume that $S_{i,j} = kl_{i,j}$, where $l_{i,j}$ is the traversed path of the

helicopter from the ship with the number $i$ to the unloading point $j$ for the transportation of cargo of mass $m$ and $k$ is the proportionality coefficient. As a result, we get

$$z_1 = \frac{k}{m} \sum_{i,j=1}^{p,n} l_{i,j} w_{i,j}. \tag{1}$$

In order to find the unloading points, we must find $min z_1$. This problem resembles the transport problem in the classical formulation, which is solved by linear programming methods [29]. Many facility location models have been made to help decision making in this area [30–36]. However, in them the solution is determined for the nodes of the spatial grid, which limits the accuracy of the optimal localization. However, such a problem cannot be solved by such methods, because the $i$ point is not a fixed point, but a point whose coordinates are only found during optimization. At the same time, we must also take into account the geographical area within which the vessels should be located. In other words, in this case the search for a solution by the methods of linear programming does not give the desired result. Usually, for transportation of goods, one vessel is sufficient, since the requirements of the delivery points do not exceed the one loaded heavy vessel. Therefore, to consider a case with more than one vessel does not make sense, since practically it is not needed. We choose a Cartesian coordinate system in which we specify the ship's coordinates $(x, y)$ and unloading points (schematically shown in Figure 3). It should be added that within the scope of the problem being solved, because of the limited distances, it is entirely permissible to abandon geographic coordinates and use Cartesian ones. Also, we limit the search for the optimal location of the ship by the boundary beyond which the vessel cannot exit—called the boundary region.

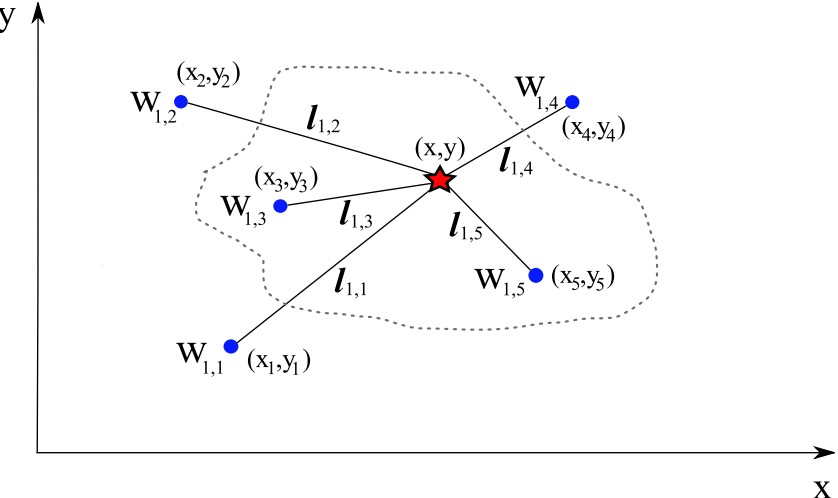

**Figure 3.** An example of the location of five points of consumption (blue circles), given by Cartesian coordinates and the best position of the ship (asterisk) for unloading by a helicopter taking into account the position constraints of the vessel (the vessel is inside the area bounded by a dashed line).

It is natural to assume that for a helicopter the minimum path from the ship to the unloading point is a straight line, and in this coordinate system $l_{1,j} = \sqrt{(x - x_j)^2 + (y - y_j)^2}$. Also, we must consider the cost of the vessel during helicopter unloading, taking into account Equation (1); as a result we obtain

$$z_2 = \left( \frac{k}{m} + \frac{Z_0}{vm} \right) \sum_{j=1}^{n} l_{1,j} w_{1,j}, \tag{2}$$

where $v$ is the speed of the helicopter and $Z_0$ is the coefficient that relates the cost of the ship's work per unit of time.

In order to find the geographic location that enables the fastest unloading (neglecting the cost), the following time function is needed

$$t_1 = \frac{1}{vm} \sum_{j=1}^{n} l_{1,j} w_{1,j}. \tag{3}$$

Let us now consider the second important factor, i.e., transportation of cargo from the point of departure of the vessel to the point of its unloading using a helicopter. In this case, the cost and time for shipping the goods, respectively, will be

$$z_3 = CL(x_0, y_0, x, y), \tag{4}$$

$$t_2 = \frac{L(x_0, y_0, x, y)}{v_0}, \tag{5}$$

where $L(x_0, y_0, x, y)$ is the distance traveled by the vessel from the starting point $(x_0, y_0)$ to the unloading point $(x, y)$ (for example, see [37]), $C$ is the coefficient connecting the costs for the carriage by the ship of the cargo per unit of traveled path and $v_0$ is the average speed of the vessel. It should be added that $L(x_0, y_0, x, y)$ is not the specified length of the ship's path to the unloading point, but the length that we are trying to find through optimization, and therefore we must establish the path of the vessel's movement.

As a result, combining both factors, we get by summing, respectively, (2) and (4), the total costs for cargo transportation by ship and helicopter will be

$$z = \left( \frac{k}{m} + \frac{Z_0}{vm} \right) \sum_{j=1}^{n} l_{1,j} w_{1,j} + CL(x_0, y_0, x, y). \tag{6}$$

Similarly, through summing Equations (3) and (5) we obtain the total time needed to transport cargo from the ship's departure point to its full unloading at its destinations,

$$t = \frac{1}{vm} \sum_{j=1}^{n} l_{1,j} w_{1,j} + \frac{L(x_0, y_0, x, y)}{v_0}. \tag{7}$$

In addition to the resulting Equations (6) and (7), it is advisable to enter a common cost function. This function will combine the cost of loading and unloading work with time costs. Let us assume that at the unloading points it is known that for a certain time interval the financial losses from the fact that there is no certain type of cargo are $F$, then the financial losses will be

$$Z = \left( \frac{k}{m} + \frac{Z_0}{vm} \right) \sum_{j=1}^{n} l_{1,j} w_{1,j} + CL(x_0, y_0, x, y) + F \left( \frac{1}{vm} \sum_{j=1}^{n} l_{1,j} w_{1,j} + \frac{L(x_0, y_0, x, y)}{v_0} \right). \tag{8}$$

Equation (8) is commonly applied to a variety of options for optimizing cargo delivery. The numerical realization of the search for the minimum of the Equation (8) is described in the Appendix A. For example, if $F$ is small and can be ignored, then Equation (8) simplifies to Equation (6). If $F$ is very large (e.g., the goods must be delivered as quickly as possible), then Equation (8) will approximate to Equation (7), and therefore have the same minimum for optimization. It should be added that if goods must be delivered as quickly as possible, hence bypassing the costs of their delivery (for example, following an environmental disaster), then it is simpler to use Equation (7), since the time is determined in an explicit form.

It is advisable to add that in the simplest case—when $L(x_0, y_0, x, y)$ is not taken into account in Equation (8) and there is no restriction on the geographical position of the vessel, the statement of the problem boils down to the well-known theory of "Standort"-

the optimal location for production, "warehouse" [38]. This problem is standardly solved by linear programming methods. In a more simplified version, when we are interested in the position of one ship between three points—the same consumers, lying at the vertices of the triangle without additional conditions, we get the Fermat-Torricelli problem with a geometrical solution in the form of a Fermat point. The problem in our formulation does not have a previously studied solution and requires additional comments on how to minimize the Equation (8), which are given in the Appendix A.

Using the proposed method, one can find the optimal place for unloading the vessel, but the question of the expediency of the applied model remains open. We must analyze how profitable the desired point for unloading ships, depending on other options for unloading. From the economic point of view, if we talk about cost, there can be another way of unloading, which will be less expensive. In this case, we need to compare the proposed method (which we will now call method No. 1) with another way of unloading. We are also interested in the case of delivery to Arctic points along the Northern Sea Route; here, an insufficient number of ports and berths, on the one hand, and shallow depths in the coastal zone and the condition of the coast, on the other, make it impossible to unload the vessel ashore without the use of a helicopter. Therefore, the only alternative to method No. 1 is the delivery of cargo by the ship from as close as possible to the point of unloading (determined by sea depth) and its unloading by helicopter at a short distance from this point and so on for other points of unloading; this is standard unloading, which here we call method No. 2. We next compare our method No. 1 with method No. 2, and to do so we must calculate the cost of unloading using both methods. In one particular scenario, method No. 1 will usually be more profitable. This would occur when the mass of the cargo being transported by the ship to its destinations $\sum_{j=1}^{n} w_{1,j} = a$ is less than some mass which here we, call critical and denote as $m_C$. In other words, the critical mass $m_C$ is the mass of cargo transported to destinations at which methods No. 1 and No. 2 give the same costs. This mass $m_C$ depends on many factors: the position and the number of unloading points, shoreline and cargo mass $p_{1,j}$ at the unloading points, as well as the parameter $F$. The calculation of $m_C$ is therefore task specific.

## 3. Results

Let us consider an example of practical importance for the delivery of goods in the area of Ob Bay, where there are significant reserves of natural gas and where the extracting and processing industry is actively developing. In this example, we demonstrate the capabilities of our method and show that the presented optimization method can have a great advantage over standard methods of cargo delivery and improve gas/oil industry logistics in this region. Let us assume that goods must be delivered to the points Tambey, Sjojaha, Antipayuta and Yamburg, all located in Ob Bay, from the point of departure of the vessel $(x_0, y_0)$ (Kara Gate), and that the ship must then go to point $(x_1, y_1)$ (see Figure 4).

The choice of the point $(x_1, y_1)$ is determined by the fact that after unloading the vessel within Ob Bay, it can continue on elsewhere without returning back to starting point. For comparison of our unloading method method No. 2 to be valid, the same departure point and final point must be used. Let us next consider the ways in which cargo can be delivered. There are restrictions on the area of the vessel's location due to the distribution of depths and the nature of the shoreline. The optimal position points of the ship are determined in each case according to the procedure described above, taking into account the safe depths and shorelines, with minimization of Equation (8). The cost calculation for our work was based on data on the average helicopter rental price, which is 2800 USD/h, and the average speed at work is which 170 km/h. The average payload capacity of a helicopter for a single journey was taken as 2 tonne (the approximate performance of the Mi-8T helicopter is taken). The ship carrying the cargo was assumed to be the NEC *Mikhail Somov*. The following data on this vessel were used: the cost per day is 14,000 USD, if the ship is worth 3000 USD per day, the average speed of the ship is 15 km/h. When using

method No. 2, we assumed that the vessel gets within an average distance of 10 km from its point of unloading.

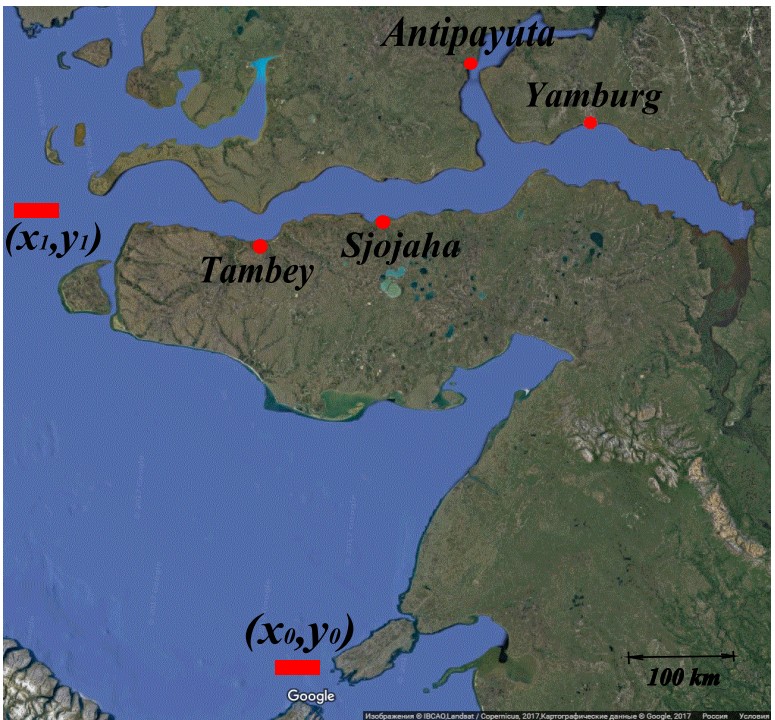

**Figure 4.** Geographical area of the example under consideration showing the unloading points.

We present the results of calculations for the critical mass, taking into account the fact that the function $F$ takes four values. The calculations are carried out using the Equation (8): we construct a graph of the function $Z/Z_S$, where $Z_S$ is the cost of delivery of the goods by the method No. 2 and $Z$ is the cost of delivering the goods by our method No. 1 (from the calculation point, i.e., minimizing function (8) as a function of the mass $m$ of the cargo carried. Figure 5 shows the function $Z/Z_S$ for five cases (see Table 1) for the carriage of goods with a mass of $m$, to destinations Sjojaha—$m_1$, Tambey—$m_2$, Antipayuta—$m_3$ and Yamburg—$m_4$. The scenario where $F \gg 1000$ in Table 1 corresponds to the fact that the final calculation is no longer dependent on the parameter $F$, i.e., with its increase, the calculation results do not change (in the case presented here, the results of the calculation cease to change noticeably as soon as $F > 10{,}000$).

**Table 1.** The values of the critical mass $m_C$ for the five cases of mass ratio and the $F$ parameter are presented, in the case of unloading the ship by helicopter to the destinations Sjojaha, Tambey, Antipayuta and Yamburg.

| | Share of Cargo $m_i/m$ Delivered to Destinations from Total Mass $m$ | | | | Critical Mass $m_C$ (tons). $F$ is set to USD/h | | | |
|---|---|---|---|---|---|---|---|---|
| Cases | Sjojaha | Tambey | Antipayuta | Yamburg | $F = 0$ | $F = 3300$ | $F = 6600$ | $F \gg 1000$ |
| 1 | 1/4 | 1/4 | 1/4 | 1/4 | 16 | 68 | 98 | 198 |
| 2 | 1/6 | 1/2 | 1/6 | 1/6 | 16 | 67 | 96 | 193 |
| 3 | 1/2 | 1/6 | 1/6 | 1/6 | 26 | 124 | 187 | 464 |
| 4 | 1/6 | 1/6 | 1/2 | 1/6 | 14 | 61 | 87 | 173 |
| 5 | 1/6 | 1/6 | 1/6 | 1/2 | 12 | 52 | 73 | 141 |

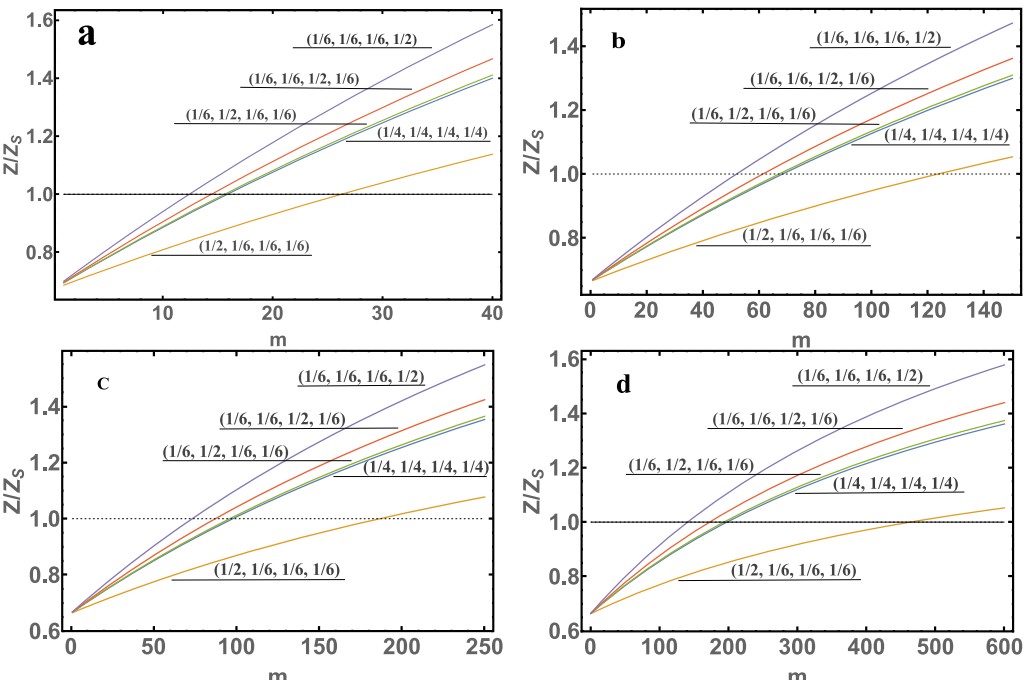

**Figure 5.** The dependence of $Z/Z_S$ for five cases (see Table 1) of the carriage of cargo of $m$ mass to the destinations is presented. The values in parentheses correspond to the values in Table 1, for example, $(1/6, 1/6, 1/6, 1/2)$ correspond to case 5 in this table. This means that $1/6$ of the total cargo carried by the vessel is unloaded at each of the discharge points of at each of the discharge points (Sjojaha, Tambey and Antipayuta), and $1/2$ is unloaded at Yamburg. (**a**) calculation for $F = 0$ USD/h, (**b**) calculation for $F = 3300$ USD/h; (**c**) calculation for $F = 6600$ USD/h; (**d**) calculation for $F \gg 1000$ USD/h.

It can be seen from Figure 5 that there exists a critical mass $m_C$. For smaller masses than $m_C$, it is advantageous to carry out unloading operations using the method presented. You can also see the benefits of unloading by our method, which can reach large values ($\sim$30%). It should be recalled that we considered an example having a concrete practical application. You can consider many other options for unloading, where the benefits can be even more significant, it all depends on the choice of the specific conditions of the problem.

Also, we will present the results of the calculation of the geographical location of the optimum unloading point of the vessel. The initial data are similar to the case considered above, see Table 1. For the weight of the load $m = 24$ tons (in the figure it is indicated by asterisks). Also considered, as a complement, the case for $F \gg 1000$ for $m = 120$ tons (in the figure it is indicated by squares). The results of the calculation are shown in Figure 6: it can be seen that the optimal unloading point could be in Ob Bay (the scenarios where $F = 0$, $F = 3300$), but may go beyond these limits and be in the Kara Sea (the scenarios where $F = 6600$, $F \gg 1000$).

It should be added that the very fact that the unloading point could be outside Ob Bay is not trivial and such options have not been considered up to now. In addition, options are possible when the unloading point at a given $F$ may be (depending on the amount of cargo at the given unloading points) both in Ob Bay and in the Kara Sea (the scenarios where $F \gg 1000$, are indicated by squares in Figure 6).

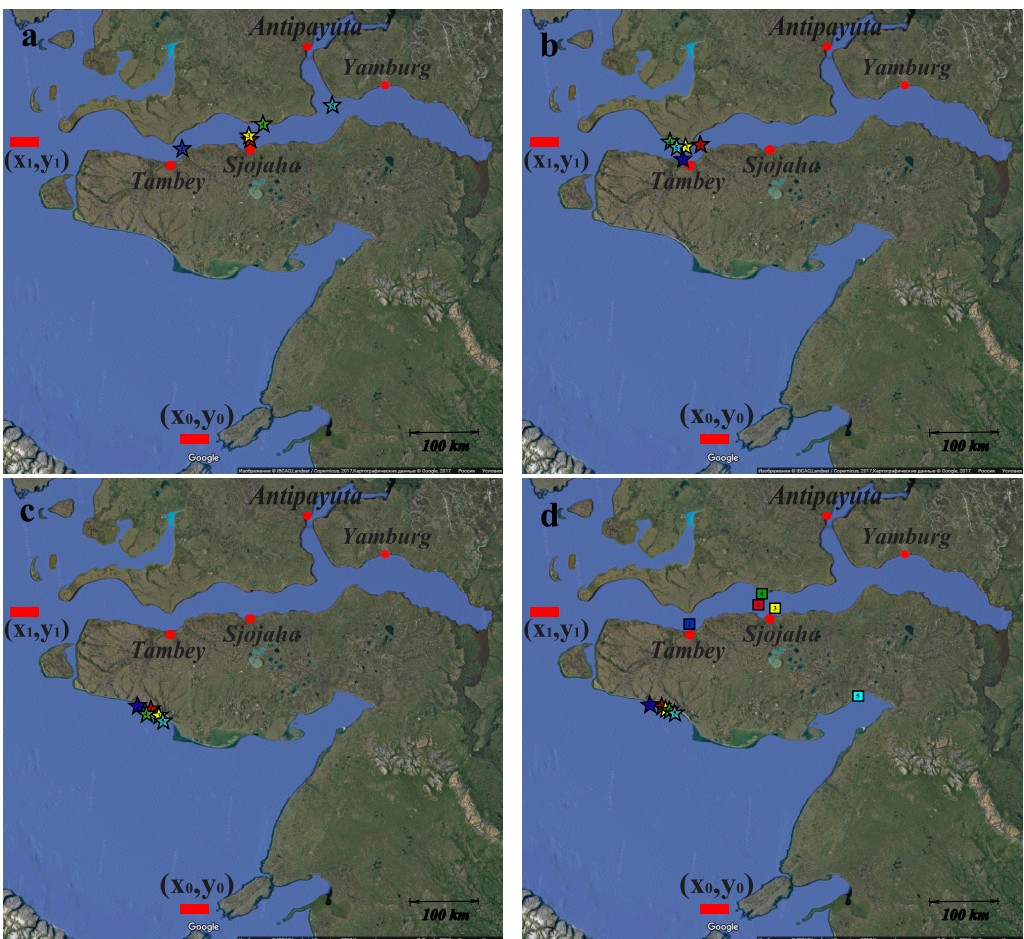

**Figure 6.** Economically optimal geographical places for unloading the vessel by helicopter are presented. The calculation corresponds to the cases presented in Table 1 for a mass of the transported cargo $m = 24$ tons: red star—1, blue star—2, yellow asterisk—3, green asterisk—4, blue asterisk—5 cases (similarly represented for mass $m = 120$ tons and denoted by squares, figure d). (**a**) calculation for $F = 0$ USD/h, (**b**) calculation for $F = 3300$ USD/h; (**c**) calculation for $F = 6600$ USD/h; (**d**) calculation for $F \gg 1000$ USD/h.

## 4. Economic Benefit Maps for Loading and Unloading Operations

Despite the fact that, it is possible to find the best geographic position for unloading a vessel using the proposed calculation method, a number of problems arise. It is not always possible to reach a given unloading place in order to unload the ship. This is due to various factors that cannot always be taken into account, for example, bad weather, shallow water, changes in the fairway, ice conditions and other circumstances affecting shipping. Moreover, there are no exact scientific approaches that enable the prediction of ice conditions [39] and climatic change [40] within the Arctic within the timescales of long-distance navigation. These obstacles can arise directly from the moment the ship moves to the point of unloading. In such situations, an flexible reaction to the current situation is necessary. Delays in decision-making due lead to financial penalties. In addition, a new unloading point must be found, but in the enfolding situation, this could be difficult. The manager who has to decide about unloading needs operational help, which can be provided quickly and in a timely manner by modeling the situation. In this case, contour maps (a map-scheme) of economic benefit will also help; these consist of the display of a geographical area, which includes the optimal point of unloading, where the economic consequences of possible points of unloading the vessel are given. As an example, we present the results of the calculations for such a map for the cases considered above(see

Figure 7): here value of $F$ = 3300 USD/h was chosen, and the calculations were carried out for cases 1–4 presented in Table 1.

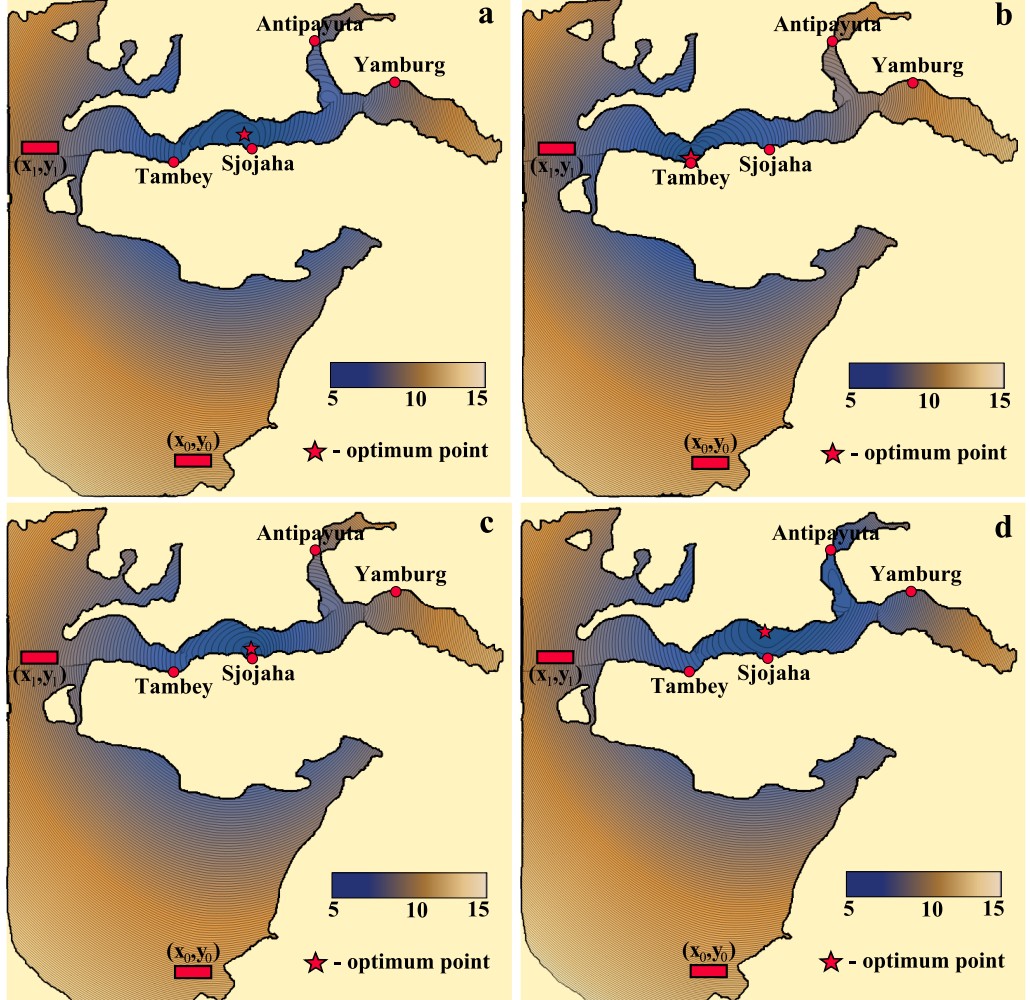

**Figure 7.** The economic benefit map for the values presented in Table 1: (**a**) case 1; (**b**) case 2; (**c**) case 3; (**d**) case 4. Calculation was carried out at $m$ = 60 tons and $F$ = 3300 USD/h. The optimum point for unloading is indicated by an asterisk. The color scale is set in the range from $5 \times 10^5$ USD to $15 \times 10^5$ USD.

Using a color scale, the drawings show where, with a minimum loss in value, a ship could be unloaded: the darker the color, the more profitable. It is also convenient and simple to analyze, and draw conclusions about, the choice of the place of discharge using isolines (black lines in Figure 7)—any point on the same isoline has the same cost of cargo delivery, despite the different geographical location of the ship's unloading point. The profit from a well-founded decision about unloading could be in the region of hundreds of thousands of dollars for one voyage. It can therefore be seen that the analysis of the search for a place to unload the ship, assuming it cannot be at the optimal point, is simple enough. Such cards would be convenient for use on a ship by the manager responsible for unloading the vessel—using them would enable he or she to determine the best place or set of places for unloading, as well as to assess the economic or time costs in each specific situation.

## 5. Conclusions

The paper presents a method of mathematical modeling of unloading and loading operations under Arctic conditions. Our method enables calculations to be performed that apply to the unloading of a vessel using a helicopter. Given that helicopters are the

principal way in which vessels are unloaded when standard operations cannot be used, our method is applicable to a range of challenging situations, and not just the Arctic.

From data presented in Figures 5–7, we can conclude that in the case of unloading a vessel in Ob Bay, the economic benefit of applying our model could be greater than 30%, but we can consider other cases where the economic benefits could be even more significant, depending on the specific geographical positions of the unloading points and the amount of unloading at these points. Qualitatively, it can be assumed that the benefit will be more significant when applied to the case of complex and lengthy shipping routes that previously had relied on delivering cargoes by using method No. 2, i.e., transporting goods directly to their destinations. This is quite obvious, because transport costs will reflect the length of time a cargo vessel has to endure Arctic conditions.

The main steps required for undertaking the mathematical modeling by the proposed method are summarized below:

1. The cost of unloading the vessel using standard methods must be calculated, i.e., method No. 2 should be used.
2. Mathematical modeling by the proposed method No. 1 should then be undertaken. Further, graphs (or tables) of the economic benefit of method No. 1 as compared with method No. 2 should be constructed (as exemplified in Figure 5).
3. If the benefit of the proposed method is substantial, a map of economic benefits should then be constructed (exemplified by Figure 7).

The presented algorithm can easily be used by any transport and logistics company for the analysis of economic benefits.

Furthermore, the benefits apply not only to the analysis of the cost of unloading and loading operations, but also in terms of time. In the event of an emergency (e.g., an environmental disaster or, threat to human life) then the cost of delivery of the necessary goods would not be an issue, because in such situations, time is paramount. In this case, it would be more expedient to use Equation (7), which will directly show the time of delivery of the cargo; however, using Equation (8) for large $F$ will yield the same result as Equation (7), when determining the optimal unloading point. It should be added that in this model, we selected specific data: the speed of the ship, the speed of the helicopter, the cost of the ship and the helicopter per unit of time. It was considered that the loading and unloading operations were proceeding without delays and that there were no force majeure circumstances. Of course, in the Arctic, all these data are approximate and may vary depending on many factors. This work can be improved by introducing different weighting factors, taking into account undefined data [41]. Despite this, even in conditions of uncertain data, using the benefit maps discussed in Section 4, it is possible to assess the profitability of a particular situation.

**Author Contributions:** Conceptualization, M.E. and D.M.; methodology, D.M.; software, D.M.; validation, M.E. and D.M.; formal analysis, M.E. and D.M.; writing—original draft preparation, D.M.; writing—review and editing, M.E. and D.M.; project administration, M.E. and D.M. Both authors have read and agreed to the published version of the manuscript.

**Funding:** This research received no external funding.

**Institutional Review Board Statement:** Not applicable.

**Informed Consent Statement:** Not applicable.

**Data Availability Statement:** Request to corresponding author of this article.

**Conflicts of Interest:** The authors declare no conflict of interest.

## Appendix A

When calculating the minimum of the Equation (8), the computer algebra system Mathematica 11.0 was used. This system allows us to find the minimum of the equation $Z$ inside the region $D$. The required region $D$ (for example, the shipping area) was obtained

by taking the coordinates with a step of 3 km along the boundary of the considered region—the Ob Bay with the Kara Sea (digitization of the geographical region).

In order to take advantage of the capabilities of the computer algebra system Mathematica 11.0, it is necessary that the Equation (8) for $Z$ be specified. In our case this is not the case, since the expression $L(x_0, y_0, x, y)$ is not specified. $L(x_0, y_0, x, y)$ is sought from the principle of the shortest path between two points between which there are geographical obstacles. In other words, this is a variational problem, which is rather difficult to solve in order to find $L(x_0, y_0, x, y)$ in explicit form in Equation (8) for $Z$. In order to find the expression for $L(x_0, y_0, x, y)$, the following algorithm was developed, based on the fact that the minimum distance between two points without geographical interference is determined by a straight line:

1. The region $D$ under consideration was divided into a set of subregion. The breakdown was conducted in such a way that within each subregion the ship could reach any point of this subset in a straight line (the line of the shortest path between 2 points if there are no obstacles).

2. First, a minimum of 1 subregion is sought; in the subregion where the ship came from. In this case, $L(x_0, y_0, x, y)$ is a straight line and is equal to $L(x_0, y_0, x, y) = \sqrt{(x - x_0)^2 + (y - y_0)^2}$.

3. Next, we search for a minimum in the next second region into which the vessel can get from region 1 moving along a straight line to the point $(x_1, y_1)$ located on the boundary of the second region. The point $(x_1, y_1)$ is chosen so that the path (straight line) from the point $(x_0, y_0)$ to the boundary of the second region is minimal. In this case, $L(x_0, y_0, x, y) = \sqrt{(x_1 - x_0)^2 + (y_1 - y_0)^2} + \sqrt{(x - x_1)^2 + (y - y_1)^2}$.

4. This procedure of finding the minimum is searched for all subregions, where in the final analysis the smallest value of all minima found from all subregions is selected.

As a result, if the minimum is in a certain region $i$, then the ship moves to this region along broken straight lines c traversed by $L(x_0, y_0, x, y) = \sqrt{(x_1 - x_0)^2 + (y_1 - y_0)^2} + \sqrt{(x_2 - x_1)^2 + (y_2 - y_1)^2} + ... + \sqrt{(x - x_i)^2 + (y - y_i)^2}$. All points $(x_1, y_1), (x_2, y_2), ..., (x_i, y_i)$ are selected in the algorithm, taking into account the shortest path from the ship's departure point $(x_0, y_0)$ to the point of its arrival $(x_i, y_i)$.

Thus setting the Equation (8), we look for its minimum using the well-known numerical minimization methods realized in the computer system of the algebra Mathematica 11.0.

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
