# Peer review of "Optimal Ways of Unloading and Loading Operations under Arctic Conditions"

_jmse, doi:10.3390/jmse9101050_

Round 1

Reviewer 1 Report

Dear authors, 

in order to improve your text some changes should be made:

all comments are in the file 

and stated as follows: 

Page 1

The abstract is very extensive, instead it should be short and consistent giving the background of the problem - first 2 sentences are enough.

then the methodology should be stated and concrete results - finding and practical solutions that may improve the existing problem

state your finding if a helicopter in your model or other possibilities of transport in your model would have achieved better results in everyday use

are there any newer findings that may support the use of drones or flying machines on auto piloting?

here the used model - example model should be shortly explained

if other models or methods were used state them too

maybe for readers non familiar with the topic to explain what is this route and where it goes - point to point - city or place

Page 2

if possible point out some cities or place on the map - or other distinct features of the route

are there different models for different type of vessels, for example by size of cargo ships - type of port - or other features ?

if so state some literature

here the authors state that different ports exist so supposing different methodologies of modeling should exist too

I'm not sure if the name should be mentioned this way, because literature is only supposed to be in numbers in brackets

Page 4 – methodology

maybe to use another phrase to start this section - improving the fluency of the paragraph

Page 7

maybe this would be one of the practical solutions to be stated in the abstract - improvement of transport for the gas/oil industry

Page 10

this part of the conclusion should be used in the abstract

this is a concrete contribution of the paper to practice

Page 12

the DOI numbers should be added where applicable

literature 26: words are together - make distance

literature 30-36 the font of the years should be bolded

Author Response

Responses to 1 reviewer's comments

1 comment:

Page 1

The abstract is very extensive, instead it should be short and consistent giving the background of the problem - first 2 sentences are enough.

then the methodology should be stated and concrete results - finding and practical solutions that may improve the existing problem

state your finding if a helicopter in your model or other possibilities of transport in your model would have achieved better results in everyday use

are there any newer findings that may support the use of drones or flying machines on auto piloting?

here the used model - example model should be shortly explained

if other models or methods were used state them too

maybe for readers non familiar with the topic to explain what is this route and where it goes - point to point - city or place

Authors' responses:

- The abstract is shortened in accordance with the comments. In this version, the abstract is less than 200 words.

- The use of this model in everyday life is not an easy question and requires additional research. Although there are many examples where this can be used (fast delivery of goods in urban environments, the use of drones, etc.). But how effective this model will be in such day-to-day use is a question for future research.

- The use of drones in this model is so far ineffective, since drones with vertical take-off and landing can take small loads and use them as an alternative to a helicopter so far too early. Although, of course, such use is very likely in the future.

- Added in the introduction:

“In January 2021, a Yamalmax-class LNG carrier for the first time made an independent transition (without icebreaker escort) from the port of Sabetta along the Northern Sea Route to the east and reached the Bering Strait. At the same time, cargo was delivered from the Yamal LNG plant to the east; the average cruising speed of the Christophe de Margerie was 9.5 knots.”

“The Ob area of the Ob Bay  is an actively developing transport highway in connection with the development of the South-Tambeyskoye oil and gas field on the Yamal Peninsula. There is a need to deliver goods to the Ob Bay region to the settlements: Sjojaha, Tambey, Antipayuta, Yamburg. At these points there are no equipped berths for the complete unloading of ships, therefore unloading is carried out by a helicopter, which is on the ship. Usually such a ship departs from Arkhangelsk. There is always the problem of choosing a point for unloading a vessel using a helicopter.”

2 comment:

Page 2

if possible point out some cities or place on the map - or other distinct features of the route

are there different models for different type of vessels, for example by size of cargo ships - type of port - or other features ?

if so state some literature

here the authors state that different ports exist so supposing different methodologies of modeling should exist too

I'm not sure if the name should be mentioned this way, because literature is only supposed to be in numbers in brackets

Authors' responses:

  • Figure 1 shows an added example (green line) presented in the introduction see "In January 2021, a Yamalmax-class LNG carrier for the first time made an independent transition ..."
  • After Eq. (1) the authors indicated the necessary literature that is used for unloading and loading operations. But this literature does not solve the problem raised in this work, for the reasons described below Eq. (1). Therefore, there was a need to develop a new approach that is presented in this work.

3 comment:

Page 4 – methodology

maybe to use another phrase to start this section - improving the fluency of the paragraph.

Authors' responses:

  • The section name has been changed to "Methods". The authors agree with the reviewer.

4 comment:

Page 7

maybe this would be one of the practical solutions to be stated in the abstract - improvement of transport for the gas/oil industry.

Authors' responses:

- Added:

“In this example, we demonstrate the capabilities of our method and show that the presented optimization method can have a great advantage over standard methods of cargo delivery and improve gas/oil industry logistics in this region.”

5 comment:

Page 10

this part of the conclusion should be used in the abstract

this is a concrete contribution of the paper to practice

Authors' responses:

- This comment has been factored into the revised abstract.

6 comment:

Page 12

the DOI numbers should be added where applicable

literature 26: words are together - make distance

literature 30-36 the font of the years should be bolded

Authors' responses:

Fixed where applicable.

Many thanks to the reviewer for the valuable comments.

Reviewer 2 Report

Searching for an optimal geographical location for unloading a vessel using helicopters is essential since this shipping method is costly and a lengthy process. Therefore, there is a need to create a support system for correct decision making in such situations. Mathematical modelling has been used to find the geographical location that ensures the most favourable and quickest cargo delivery from a vessel to its destination using a helicopter. A criterion has also been found in which the search for the optimum point is a more rational way of unloading the vessel than other discharge options. It is shown that significant economic and temporal benefits can be gained when delivering goods to hard-to-reach points during the transportation of goods in Ob Bay. The developed model can be extended to the case of delivery of cargoes under Arctic conditions and other places where transport infrastructure is underdeveloped. The paper is well written scientific work. However, some minor improvements should be made. My comments are as follows:

  • language corrections should be made;
  • it should be possible to use uncertain data in the evaluation process (fuzzy sets, interval data, hesitant fuzzy sets, Z-numbers, D-numbers and so on). Please comment widely in the manuscript. E.g. 'D Numbers–fucom–fuzzy Rafsi Model For Selecting The Group Of Construction Machines For Enabling Mobility' etc.
  • Further research directions should be extended in the conclusions section;

Author Response

Responses to 2 reviewer's comments

comments:

Searching for an optimal geographical location for unloading a vessel using helicopters is essential since this shipping method is costly and a lengthy process. Therefore, there is a need to create a support system for correct decision making in such situations. Mathematical modelling has been used to find the geographical location that ensures the most favourable and quickest cargo delivery from a vessel to its destination using a helicopter. A criterion has also been found in which the search for the optimum point is a more rational way of unloading the vessel than other discharge options. It is shown that significant economic and temporal benefits can be gained when delivering goods to hard-to-reach points during the transportation of goods in Ob Bay. The developed model can be extended to the case of delivery of cargoes under Arctic conditions and other places where transport infrastructure is underdeveloped. The paper is well written scientific work. However, some minor improvements should be made. My comments are as follows:

  • language corrections should be made;
  • it should be possible to use uncertain data in the evaluation process (fuzzy sets, interval data, hesitant fuzzy sets, Z-numbers, D-numbers and so on). Please comment widely in the manuscript. E.g. 'D Numbers–fucom–fuzzy Rafsi Model For Selecting The Group Of Construction Machines For Enabling Mobility' etc.
  • Further research directions should be extended in the conclusions section;

Authors' responses:

  • English is improved in this version of the manuscript.
  • In the conclusion, the answer is immediately given to the last two comments:

“It should be added that in this model, we selected specific data: the speed of the ship, the speed of the helicopter, the cost of the ship and the helicopter per unit of time. It was considered that the loading and unloading operations were proceeding without delays and that there were no force majeure circumstances. Of course, in the Arctic, all these data are approximate and may vary depending on many factors. This work can be improved by introducing different weighting factors, taking into account undefined data, eg [41]. Despite this, even in conditions of uncertain data, using the benefit maps discussed in Section 4, it is possible to assess the profitability of a particular situation.”

Many thanks to the reviewer for the valuable comments.
